# Origin of Room-Temperature Ferromagnetism in Hydrogenated Epitaxial Graphene on Silicon Carbide

**DOI:** 10.3390/nano9020228

**Published:** 2019-02-08

**Authors:** Mohamed Ridene, Ameneh Najafi, Kees Flipse

**Affiliations:** Molecular Materials and Nanosystems, Eindhoven University of Technology, 5600 MB Eindhoven, The Netherlands; ridane_m@outlook.com (M.R); ameneh.na@gmail.com (A.N.)

**Keywords:** hydrogenated epitaxial graphene, electronic structure, ferromagnetism

## Abstract

The discovery of room-temperature ferromagnetism of hydrogenated epitaxial graphene on silicon carbide challenges for a fundamental understanding of this long-range phenomenon. Carbon allotropes with their dispersive electron states at the Fermi level and a small spin-orbit coupling are not an obvious candidate for ferromagnetism. Here we show that the origin of ferromagnetism in hydrogenated epitaxial graphene with a relatively high Curie temperature (>300 K) lies in the formation of curved specific carbon site regions in the graphene layer, induced by the underlying Si-dangling bonds and by the hydrogen bonding. Hydrogen adsorption is therefore more favourable at only one sublattice site, resulting in a localized state at the Fermi energy that can be attributed to a pseudo-Landau level splitting. This *n* = 0 level forms a spin-polarized narrow band at the Fermi energy leading to a high Curie temperature and larger magnetic moment can be achieved due to the presence of Si dangling bonds underneath the hydrogenated graphene layer.

## 1. Introduction

In the last decade, chemical functionalization of graphene attracted a good deal of attention among scientists for various reasons [1,2,3,4,5,6,7,8,9,10,11,12,13,14,15,16,17,18,19,20,21,22,23]. The common motivation is the modification of electronic properties of graphene. The electronic properties are much more affected under the influence of covalent functionalization, which encourages researchers to focus on studying the decoration of various defects in graphene [3,5,6,10,13,15,16]. In this way, it is also possible to make graphene spintronic devices with the aim of utilizing spintronics technology [3,4,6,8,9,10,12,15,22]. An important issue in exploring graphene-based spintronics is how to introduce magnetic order in graphene and in particular ferromagnetism. In this approach, ferromagnetism was predicted theoretically for hydrogenated graphene [4,18,24]. For example, using spin-polarized density functional theory (DFT), Zhou *et al.* [17] have shown that semihydrogenated graphene, a layer of graphene in which half of the carbon atoms (all belongs to one sublattice) are bonded to an hydrogen atom, is a ferromagnet with an estimated Curie temperature between 278 K and 417 K. The suggested mechanism is the coupling of unpaired localized electrons, present at non-hydrogenated carbon atoms due to the breaking of the π bonding network of graphene. Their calculations show that a ferromagnetic configuration is the ground state of such a system. In principle, defects are associated with lattice distortions, vacancies or chemical functionalization, introducing localized states at the Fermi level in graphene [2]. Onsite Coulomb electron–electron interactions lead to spin-polarization of these dispersionless flat states at the Fermi level and thus the existence of local magnetic moments. The magnetic properties of graphene are determined by the interaction of the mentioned localized states. In the literature, several different mechanisms have been proposed for ferromagnetism of graphene. In Ref. [4] the authors calculated the coupling of defect-induced extended states, assuming two different types of defects, the hydrogen chemisorption defect and the vacancy defect by first principles electronic structure calculations, showing either ferromagnetic or antiferromagnetic ordering depending on whether the defects are distributed on the same or different sublattices of the graphene lattice. In this last approach, room-temperature ferromagnetism is explored theoretically for hydrogenated graphene, but experimentally it was only obtained for epitaxial graphene on silicon carbide (SiC) [25]. This was the motivation to study the interplay between the structural properties and the electronic structure of hydrogenated epitaxial graphene.

We will show that in the case of epitaxial graphene specifically on SiC, the presence of the Si-dangling bonds creates preferential sites for hydrogen adsorption [26]. This leads to a preferred sublattice carbon occupation of H atoms, which is necessary for ferromagnetic behaviour. Importantly, our ab-initio calculations show that the created localized state at the Fermi energy can be attributed to a pseudo-Landau level splitting. This *n* = 0 level forms a spin-polarized narrow band at the Fermi energy leading to a high Curie temperature and larger magnetic moment can be achieved due to the presence of Si dangling bonds underneath the hydrogenated graphene layer. Hence, at first the structural properties of epitaxial graphene on SiC and the effect of hydrogen chemisorption on magnetic ordering will be described, followed by our ab-initio calculations of the electronic structure of the system will be discussed.

## 2. Materials and Methods

Ab-initio calculations were performed based on the density functional theory (DFT) within the generalized gradient approximation (GGA) as implemented in the SIESTA computational code [27] within the Perdew-Burke-Ernzerhof (PBE) form [28]. The core electrons were replaced by Troulier-Martins pseudopotentials [29]. A double-ζ basis set of localized atomic orbitals was used for the valence electrons. Sampling of the Brillouin zone has been performed using by a (10×10×1) shifted Monhorst-Pack grid [30], while a mesh cut-off energy of 200 Ry has been imposed for real-space integration. All structures have been relaxed until forces were less than 0.05 eV/ Å. The influence of the SiC substrate was introduced by an extra Hubbard-U term to account for electron correlation as a result of the Coulomb interaction between the Si localized electrons [31]. Therefore, this correction which is an effective potential defined by U_eff_ = U − J, where U and J are the Coulomb and the exchange parameters [32] are only applied to the 3p orbitals of the Si-dangling bonds. In the calculations, U_eff_ is considered equal to 3 eV which is in a good agreement with the experimental results [31]. A vacuum interval of more than 10 Å was used to avoid the interaction between the periodic slabs.

## 3. Results and Discussion

### 3.1. Preferential Hydrogen Adsorption Sites in Epitaxial Graphene on SiC(0001)

Upon annealing SiC at high temperature, growing graphene on the Si face of 6H-SiC substrate, a first carbon layer strongly bonded to the substrate is formed. This layer does not present the electronic properties of graphene since it is interacting with the substrate by means of the covalent bonds between the C atoms of this layer and the Si atoms of the substrate surface. However not all the C atoms bond to the Si atoms, so there are some Si-dangling bonds existing underneath. This layer is well known as the buffer layer. The first layer of the C sheet on top of the buffer layer is considered as the first layer of graphene which shows the linear dispersion relation at the Dirac point. Characteristic for scanning tunneling microscopy (STM) images of epitaxial graphene on SiC(0001) is the observed Moiré pattern due to the presence of the buffer layer below the graphene layer [33]. Instead of a (6√3×6√3)R30° reconstruction, expected from a reconstruction point of view with respect to SiC lattice and observed in low energy electron diffraction (LEED) pattern of epitaxial graphene [34], STM images show only a superstructure with a quasi − (6×6) SiC periodicity (∼ 1.8 nm) due to electron interference effects [see Figure 2(d)] [35]. This superstructure is related to the existence of intrinsic curved regions in epitaxial graphene [35]. Ab-initio calculations have shown that the Si-dangling bonds in the buffer layer induce these curved regions and, consequently, the graphene layer mimics this morphology to form a quasi—(6×6)SiC pattern of bright spots observed in the STM image (Figure 2d). To clarify that, the total charge density of the buffer layer states is shown in Figure 1a–c. As can be seen from Figure 1(a), an apparent quasi − (6×6)SiC modulation, denoted by the full line diamond cell, is recognized in the total charge density of buffer layer states.

Patterns with dark regions are separated from each other by brighter borders. The dark regions represent the C atoms which are covalently bounded to the Si atoms of the substrate and the bright spots are associated with the C atoms with no interaction with the substrate, so leaving the Si atom with a dangling bond underneath. Figure 1b shows a few of these irregular hexagonal shapes and the cross-section of those is depicted in Figure 1c. Here one can see the curving of the surface due to the existence of Si-dangling bond. These convex regions are mimicked by the first layer of graphene, as shown in Figure 2a.

Therefore, the total charge modulation density of graphene layer states exhibits a honeycomb lattice plane containing corrugations [see cross section in Figure 2c] with the quasi − (6×6)SiC periodicity. The corrugations in graphene layer are smoother than those of the buffer layer. There is some evidence which indicates that these convex regions are favourable sites for hydrogen adsorption [26]. STM images of an epitaxial graphene on SiC before and after hydrogenation are displayed in Figure 3. According to Ref. [26], hydrogenation is performed with a very low coverage of 0.76 ± 0.17% of the surface. Comparing Figure 3a,b, they clearly show protrusions at the corners of (6×6)SiC superstructures after hydrogenation process. This observation indicates that hydrogen atoms preferentially attach to convexly curved regions of monolayer graphene grown on SiC(0001) but not to the concave areas.

Thereafter, the sample is heated and it is shown that above ∼630 °C, hydrogen atoms start to desorb from the surface, and as a result the height of the protrusions decreases and at the point where the sample is annealed up to 680 °C, the corrugations came back to the levels of pristine graphene. A (6×6)SiC superstructure with respect to SiC surface has a periodicity of ∼1.8 nm, corresponding to ∼7.5 times the lattice parameter of graphene (0.24 nm), and recent theoretical work has shown that hydrogen adsorption is more favourable at only one sublattice (A or B) with sublattice distances larger than 1 nm [36]. Since magnetic moments arise from this hydrogen adsorption on graphene (see below), an imbalance of occupied sublattice sites will lead to a ferromagnetic ground state obeying the Lieb theorem [37]. This defines the ground state total spin angular momentum of the system with a bipartite lattice and a half-filled band based on the difference in the number of sites in the A and B sublattices, i.e. S=1/2||A|−|B||, yielding itinerant-electron ferromagnetism with |A|≠|B|. In the case of hydrogenated graphene, the difference is between the occupation of A or B sublattice sites with hydrogen. In epitaxial graphene the Si-dangling bond induced curving of the graphene layer creates a superlattice for hydrogen adsorption, preventing a random occupation of both sublattices which would be the case in hydrogenated quasi-free standing epitaxial graphene that does not show a ferromagnetic response [25].

### 3.2. Room-Temperature Ferromagnetism in Hydrogenated Epitaxial Graphene on SiC(0001)

We performed electronic structure calculations on hydrogenated graphene on SiC based on density functional theory (DFT) calculations using the SIESTA code. In first instance order the influence of hydrogen chemisorption on curved graphene was studied. The curved graphene layer with H adsorbed system was relaxed, so the initial curvature is not necessarily kept. An effective electron doping of the graphene layer, as is the case for graphene on SiC, was simulated by a shift of the Dirac point of 0.5 eV below the Fermi Energy; after adding a hydrogen atom and keeping the same value of the doping, the state is shifted towards the Fermi energy by ~0.3 eV, as is shown in Figure 4b. This indicates a possible hydrogen induced p type doping. The role of SiC and the buffer layer will be considered explicitely in Section 3.3. We considered the graphene layer as a curved layer to retain the induced rippling with a (6×6)SiC superstructure of the underlying buffer layer. In order to simulate the quasi − (6×6)SiC pattern, a curved (7×7)_G_ cell containing 98 carbon atoms was applied (the subscript ”G” is used to indicate that the lattice parameter is seven times the lattice parameter of graphene, i.e. 0.24 nm). This is a reasonable choice for a DFT calculation of the electronic structure since it has a periodicity close to the quasi − (6×6) SiC periodicity observed in STM images of epitaxial graphene on SiC (Figure 2d). According to the observations of Ref. [26], hydrogen atoms preferentially bond to sites where the lattice is maximally convexly curved, which enhances the transition from sp^2^ to sp^3^. Therefore, we added one hydrogen atom to the highest carbon atom in the lattice unit cell of calculations. The relaxed structure of the curved (7×7)G graphene cell with one H adatom is shown in Figure 4a.

The adsorbed H atom breaks a π bond and slightly pulls out the carbon atom below due to a covalent bond interaction. This carbon atom changes its hybridization state from sp^2^ to sp^3^ and the overall structure changes, causing a structural modulation, as was discussed in Reference [4]. The corresponding DOS (black solid line in Figure 4b) exhibits relatively intense peaks compared to the DOS of the curved (7×7)G (red dot line). The state at the Fermi energy (which is the position of the Dirac point [E_D_]) has been obtained and reported by previous work [4] as arising from the carbon hydrogen bond, but the presence of states symmetrically located around it reflects another nature. Interestingly, similar as in Ref. [38], the contributing peaks can be fitted to a series of Landau levels in graphene (Figure 4c), satisfying the relation: (1)$$E_{n}=E_{D}\pm \sqrt{2ev_{F}\hbar B_{s}\left|n\right|}$$
where, *E_D_* is Dirac energy, e is elementary charge, *ν_F_* is Fermi velocity, ℏ is reduced Planck constant (h/2π) and B_s_ is a pseudo-magnetic field. This indicates that the observed peaks in the DOS can be assigned to pseudo-Landau levels (PLLs). It is worth to note that the H-induced PLLs are not only caused by ”mechanical” strain (as is clear from red dashed line of Figure 4b which belongs to curved graphene cell) but the redistribution of the hopping integral values is also changed by the chemical change from sp^2^ to sp^3^ in a similar way as for strained graphene with a triangular symmetry [38,39]. As interpreted in Refs. [40,41], structural modulations cause a perturbation in the hopping parameter such as t→t+δt for a C atom experiencing strain. As a result, the energy eigenvalues of the system change with the variation of the wave vector $\vec{k}\to \vec{k}+\delta \vec{k.}$ In analogy to a 2D epitaxial graphene submitted to a perpendicular magnetic field where the momenta of the electrons are shifted like $\vec{k}\to \vec{k}+\left(e/h\right)\vec{A}$ ($\vec{A}$ is the vector potential), the modulation in the hopping parameter t due to the C-H sp^3^ hybridization induces an effective vector potential (pseudo-vector potential) and, hence, a pseudo-magnetic field. Using a Fermi velocity of 1.0 × 10^6^ ms^−1^, the resulting pseudo-magnetic field extracted from the fit is ∼250 T and its value increases for higher hydrogen coverage and vice versa. This is in agreement with previous theoretical results [39].

From the value of the pseudo-magnetic field, it is noteworthy that the magnetic length ($l_{B}≅26/\sqrt{B\left[T\right]}$nm) is comparable to the distance between the hydrogen atoms, i.e. the unit cell used in the calculations. This is important since it indicates that the entire region between hydrogen atoms is affected by the magnetic field. Nevertheless, the dependence of the magnetic field magnitude on the cell leads to an extremely inhomogeneous magnetic field distribution on the surface that contributes to the localization of graphene quasiparticles [42]. By including exchange interactions between C atom and H atoms into the calculations, similar as what was shown by Yazyev and Helm [4], the half-filled localized state at the Fermi level (PLL *n* = 0) will split into two states; filled state (spin up) and empty state (spin down) denoted by red and blue lines in Figure 4d, respectively. In this case, the spin-splitting is ∼200 meV corresponding to the coverage of ∼1%, which is expected to increase for higher coverage [4]. So, we can concluded that the exchange interaction, which leads to ferromagnetic behavior, arises from the C–H chemisorption induced pseudo-Landau level states at the Fermi level in hydrogenated graphene. According to Lieb’s theorem [37], the condition to have magnetic order is an unequal occupation of A and B sublattices by hydrogen. This statement is clearly illustrated in Figure 5.

The exchange interaction between localized electron states leads to a texture of magnetic states on the H site and the odd nearest neighbours of the carbon atom below H, which exhibits a triangular symmetry and extends to similar regions around the H atoms. The preference for H adsorption on certain sites in epitaxial graphene on SiC is due to the existence of Si-dangling bonds underneath, i.e. the corners of the quasi − (6×6) SiC superstructure. Our calculation results for the isosurface of the spin density distribution of four hydrogen adsorbed atoms, occupying an A sublattice at the corners of the quasi – (6×6) SiC superstructure, are shown in Figure 6a.

The attached hydrogen atoms (remarked by arrows in the figure) are separated from each other by ∼1.7 nm, and, as clearly seen, the localized electron states extend over a few graphene unit cells around the H atoms. The long-range magnetic coupling is therefore attributed to these connected regions of spin-polarized quasi-localized states.

Very unexpectedly, experimental results of hydrogenated epitaxial graphene show room- temperature ferromagnetism [25,43]. However, s-p electron systems can generate ferromagnetism with relatively high Curie temperatures if a narrow band of localized states is formed in a gap, or a pseudo-gap, which leads to a high value of magnon stiffness so that the low-lying spin fluctuations do not reduce the critical ordering temperature in such systems [44]. The (spin-polarized) band structure of hydrogenated graphene is shown in Figure 6b. It is obvious that the hydrogenation process leads to the disappearence of the linear dispersion of pristine graphene at the K point and opens a gap. In addition, the PLL *n* = 0 gives rise to two exchange split states lying in a band gap which is defined by the energy difference between the PLLs *n* = 1 and *n* = −1. The bandwidth of the filled (spin up) band is ~0.1 eV and the empty (spin down) band is almost flat. This is in agreement with Ref. [44], therefore, the Curie temperature can be deduced from the exchange splitting of the PLL *n* = 0 which is ~200 meV. This value is well above k_B_T (k_B_ is the Boltzmann constant) at room temperature.

### 3.3. Substrate Effect on the Electronic Properties of Hydrogenated Epitaxial Graphene

In the previous sections, the origin of the room-temperature ferromagnetic property of hydrogenated graphene on SiC was, for simplicity, discussed excluding the effect of the SiC substrate and buffer layer in the electronic structure calculations, just to indicate the connection between the role of local strain and the existence of pseudo-Landau levels. So far, a ferromagnetic behaviour is reported only for hydrogenated graphene, which is grown on a SiC substrate [25,43]. So it is important to include the effect of SiC as well as buffer layer, explicitly, in the calculations to find out their specific role. For the most realistic calculation a (6√3×6√3)R30° unit cell with respect to SiC should be taken into account, which means a cell of (13×13) graphene units. This is a large cell and a time-consuming calculation, therefore it is very common in calculations to replace the (6√3×6√3)R30° cell periodicity with a (√3×√3)R30° SiC cell which contains (2×2) graphene units [45,46]. In order to get commensurability between a (2×2) graphene cell and a (√3×√3) SiC cell rotated by 30 degrees, stretching of the graphene layer by 8% is needed. Adsorbing a hydrogen atom will overestimate the C-C bond, which will break for sp^2^ to sp^3^ hybridization. In order to avoid this, we compress the (√3×√3)R30◦ SiC by 7% instead of stretching it. This leads to a very high coverage of hydrogen (∼12%) atoms in the unit cell and therefore it was replaced by a (4×4) graphene cell, commensurate with a (2√3×2√3)R30◦ SiC substrate, while keeping the 7% compression of the substrate to get the commensurability between the two lattices. This was also used for epitaxial two-layer graphene on the Si-face of SiC by Gao and Tosatti [47]. Adding one H atom to this unit cell results in a ∼3% hydrogen coverage. The relaxed structure in the presence of H adatom is shown in Figure 7. It is worth mentioning that the geometry relaxation took place in different steps. Firstly, the SiC layers were saturated by hydrogen atoms on the C-face (bottom face) and were relaxed until forces were less than 0.05 eV/Å. Secondly, the first graphene layer is added on top and the structure was free to relax except the bottom SiC layer; this led to the buffer layer formation. The second graphene layer is then added and relaxed and finally the hydrogen atom is added on top of the highest carbon atom of the graphene layer (which coincides with the position of the Si atom with a dangling bond in the buffer layer); as previously mentioned, all the atoms were free to move except the bottom SiC layer and the structure is relaxed until forces were less than 0.05 eV/Å.

From the side view of the structure in Figure 7a, it is clear that two compressed bilayers of 6H-SiC were used with the dangling bonds on the bottom C-face saturated with the hydrogen atoms. On top of SiC layers, a (4×4) graphene honeycomb lattice is added to model the buffer layer, and then a second (4×4) graphene lattice is added in a Bernal stacking with the first one to form the epitaxial graphene layer. The curving of the graphene layer is not so clear due to the small cell that we used. Ab-initio calculations were performed based on the density functional theory (DFT) within the generalized gradient approximation (GGA) as implemented in the SIESTA computational code within the Perdew–Burke–Ernzerhof (PBE) form.

The spin unrestricted calculated band structure of epitaxial graphene and hydrogenated epitaxial graphene on SiC are shown in Figure 8a,b, respectively. In Figure 8a, the n-doped characteristic of epitaxial graphene on SiC is clearly evidenced by the position of Dirac point shifted to ∼0.5 eV below the Fermi level at the K point, this is consistent with the previous experimental and theoretical results [34,46]. However, unlike other results [45], our calculations result in a spin-polarized band structure for epitaxial graphene. This could be interpreted as a side effect of the defined unit cell in the calculations in which the SiC substrate is compressed.

However, it is very likely to be attributed to Coulomb interactions between the localized electrons of the Si-dangling bonds. Looking more closely, while no spin polarization was obtained for the graphene π-electrons, each localized state attributed to the Si-dangling bonds [45,46] splits into two spin-polarized states, one filled and one empty. This may explain the paramagnetic response of the buffer layer ascribed to the presence of the Si-dangling bond localized states in the buffer layer [25], resulting in a total magnetic moment of ∼4μB per unit cell considered in the calculations. Comparing the spin-polarized band structure of hydrogenated epitaxial graphene (Figure 8b) with epitaxial graphene (Figure 8a) shows notable differences. It is shown that the linear dispersion relation disappears at the Dirac point. Additional narrow spin-polarized bands (indicated by arrows in Figure 8b) appear besides the remaining intact (or with a small position change) spin-polarized bands of the Si-dangling bonds. These two extra spin-polarized bands correspond to the split PLL *n* = 0. In the geometry we utilized, the hydrogen coverage is ∼3%; this leads to the observation of a filled (majority spin) pseudo-Level *n* = 0 at about 0.3 eV below the Fermi energy at the K point. This value may decrease if the hydrogen coverage is less and vice versa [4]. As stated above, the energy ”gap” between the PLLs n = 0 and *n* = −1 decreases when the hydrogen coverage is reduced since it is related to the pseudo-magnetic field.

In addition, the calculated total magnetic moment without the SiC nor the buffer layer contribution but with the simulated effective electron doping as discussed in 3.2, is 0.5 μB per H adatom. This value is smaller than the one obtained when the Si-dangling bonds of the buffer layer are present in 3.3, which is 0.7 μB per H adatom. This indicates that Coulomb interactions between localized Si-dangling bond states and the quasi-localized PLL *n* = 0 could play a role leading to an increase in the exchange splitting of this latter and hence gives rise to a larger magnetic moment.

## 4. Conclusion

In conclusion, our electronic structure calculations in combination with published experimental results provides a coherent picture to understand the observed room-temperature ferromagnetism in hydrogenated epitaxial graphene on SiC. We have theoretically shown that specific carbon sites, due to the presence of Si-dangling bonds in the underlying buffer layer, result in a quasi (6×6)SiC superstructure of epitaxial graphene, favouring one sublattice site hydrogen occupation. The chemical modification of the graphene layer by H adatoms and strain-like effects generate a pseudo-magnetic field and consequently localized pseudo-Landau levels in the electron density of states. Extended spin-polarized electron states from the pseudo-Landau level *n* = 0 give rise to long-range ferromagnetic coupling. The relative high Curie temperature is obtained due the narrow bandwidth of these states formed in a bulk energy gap leading to a high magnon stiffness. Furthermore, Coulomb interactions between the pseudo-Landau level and the Si-dangling bond states can lead to an increase of the total magnetic moment. The presence of localized electron states close to the Fermi level is evidenced by photoemission spectroscopy measurements, which also indicates the p-doping of epitaxial graphene due to hydrogenation.

## Figures and Tables

**Figure 1 nanomaterials-09-00228-f001:**
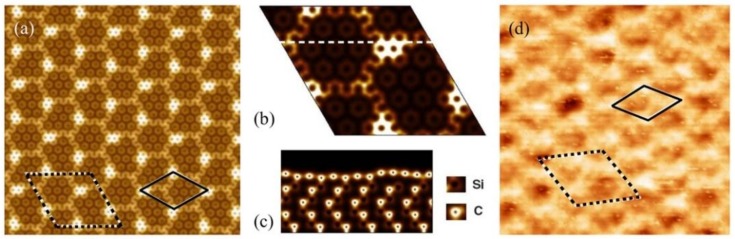
Buffer layer structure on top of the Si-terminated SiC surface (reprinted figure from Ref. [35], with the permission of APS Publishing). (**a**) 11 × 11 nm^2^ image of the total ab-initio charge density; the common (6√3×6√3)R30° SiC unit cell is shown by the dashed line diamond cell and the incommensurate (6×6) SiC modulation is denoted by full line diamond cell; (**b**) The irregular hexagonal patterns inside a (6√3×6√3)R30° SiC unit cell with a cross section along the defined dashed line shown in (**c**); (**d**) 12 × 12 nm^2^ STM image of the buffer layer (V = −2.0 V and I = 0.5 nA); the superstructure (6×6)SiC is clearly visible.

**Figure 2 nanomaterials-09-00228-f002:**
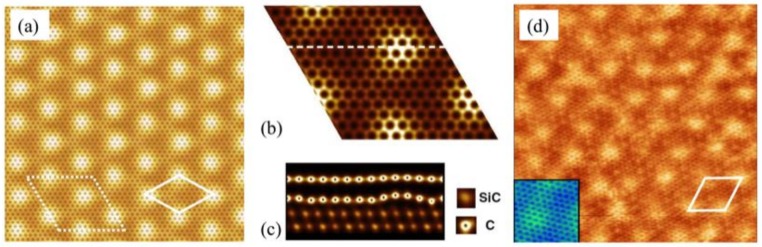
The first graphene layer on top of the buffer layer of Si-terminated SiC surface (reprinted figure from Ref. [35], with the permission of APS Publishing). (**a**) 11 × 11 nm^2^
image of the total ab-initio charge density; the common (6√3×6√3)R30° SiC unit
cell and the incommensurate (6×6)SiC modulation are defined by a dashed line
and a full line diamond cell respectively; (**b**) The total charge density
inside the (6√3×6√3)R30° SiC unit cell; (**c**) The cross section of the
dashed line defined in (b); (**d**) 12 × 12 nm^2^ STM image of the
monolayer graphene (V= −0.2 Vand I= 0.1 nA); the periodicity of (6×6)SiC is
indicated by a full line.

**Figure 3 nanomaterials-09-00228-f003:**
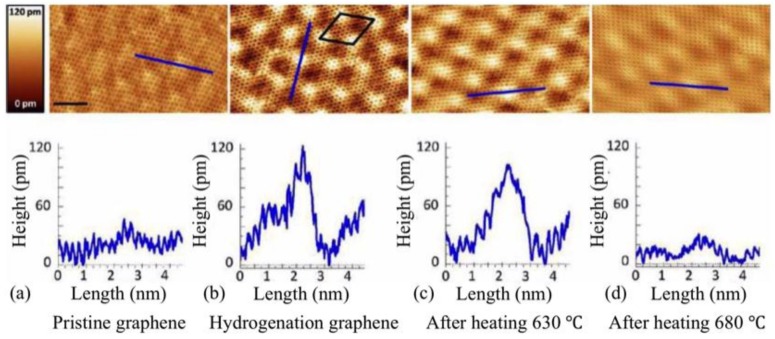
Room-temperature STM images of pristine monolayer graphene and hydrogenated graphene on Si terminated SiC (reprinted figure from Ref. [26], with the permission of ACS Publishing). (**a**) Pristine graphene (V = 115 mV and I = 0.3 nA); (**b**) hydrogenated graphene with a low coverage of 0.76±0.17% (V = 50 mV and I = 0.3 nA); (**c**) hydrogenated graphene after annealing for 5 min at 630 °C (V = 50 mV and I = 0.3 nA); (**d**) hydrogenated graphene after annealing for 5 min at 680 °C (V = 50 mV and I = 0.3 nA). The cross sections, shown below each STM image, indicate the increase of corrugation at the corners of the quasi − (6×6) superstructure after hydrogenation and decreasing of them by annealing the hydrogenated graphene due to desorption of hydrogen atoms. Scale bar is 2 nm and the sizes of images as well as the cross section lines are the same.

**Figure 4 nanomaterials-09-00228-f004:**
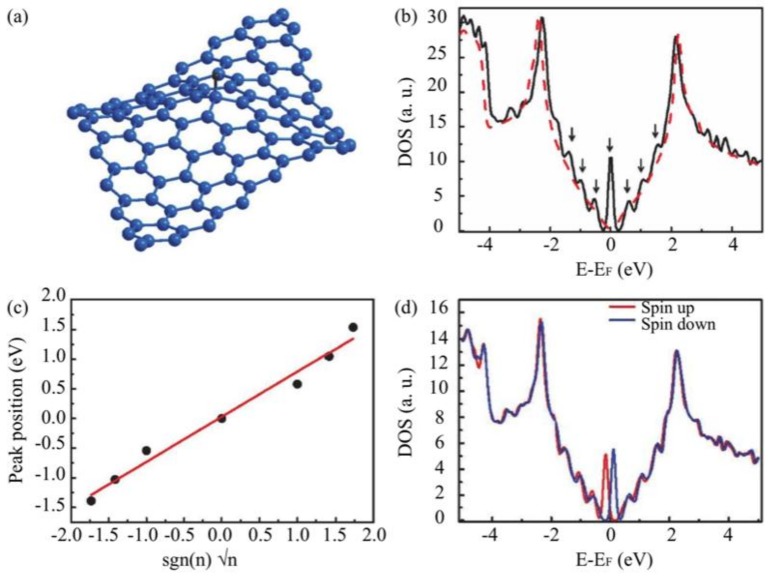
Pseudo-Landau levels in hydrogenated epitaxial graphene. (**a**) Relaxed curved (7×7) graphene cell upon hydrogen adsorption; (**b**) The total density of states of the structure in (a) (black line) which exhibit changes compared to the total DOS of curved graphene (red dashed line); (**c**) Fitting of the localized state appearing at the Dirac point and other states symmetric to it using the Landau levels dispersion in grapheme; (**d**) Spin-polarized DOS showing the pseudo-Landau levels exchanged split.

**Figure 5 nanomaterials-09-00228-f005:**
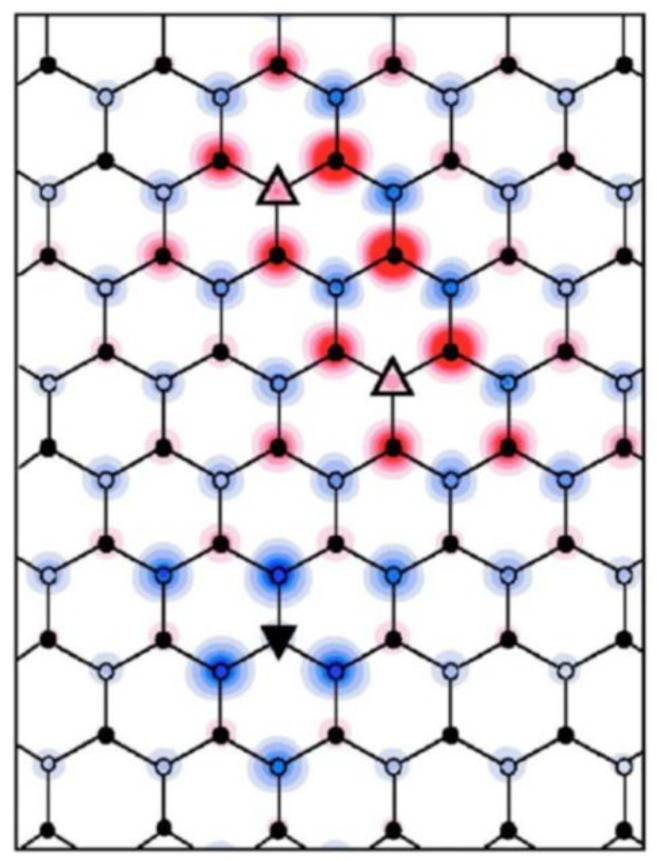
Spin density distribution of hydrogenated graphene with two hydrogen attachments on sublattice A and one hydrogen chemisorption on sublattice B of graphene (reprinted figure from Ref. [4] with the permission of APS Publishing). Red and blue represents opposite spin directions.

**Figure 6 nanomaterials-09-00228-f006:**
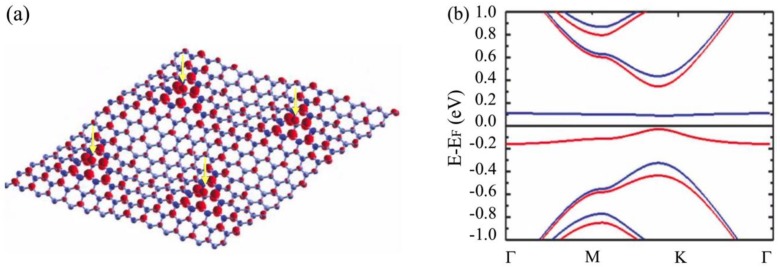
(**a**) Isosurface representation of the spin density distribution around four hydrogen adatoms on sublattice A sites (indicated by arrows); (**b**) Spin-polarized band structure of hydrogenated (7×7) graphene cell. Spin-up bands are in red and spin-down bands are in blue.

**Figure 7 nanomaterials-09-00228-f007:**
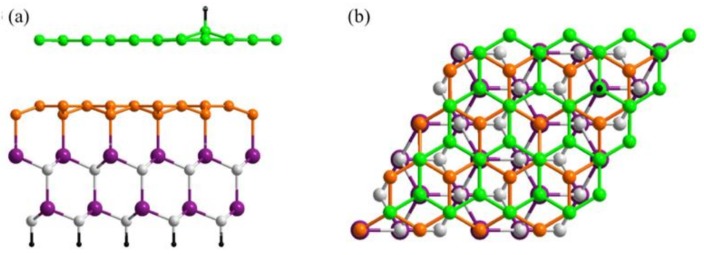
Ball and stick images of hydrogenated Si-face epitaxial graphene with 3% hydrogen coverage. (**a**) Side view and (**b**) top view of hydrogenated epitaxial graphene. The ((2√3×2√3)R30° SiC substrate (violet for Si atoms and gray for carbon atoms) is saturated by hydrogen atoms (black) and compressed to host a (4×4) graphene cell to model the buffer layer (orange atoms). The epitaxial graphene layer (green atoms) is in a Bernal stacking with the buffer layer and has the same dimensions (4×4).

**Figure 8 nanomaterials-09-00228-f008:**
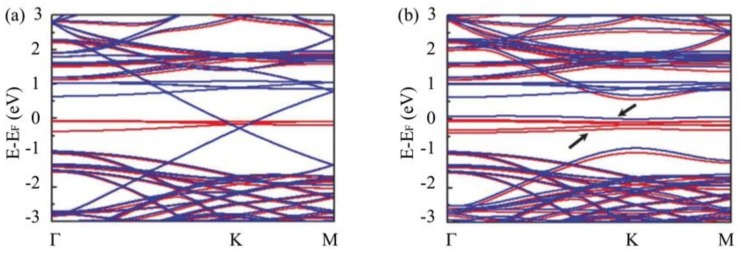
Influence of the SiC substrate and the buffer layer on the band structure of epitaxial graphene and hydrogenated epitaxial graphene. (**a**) Spin-polarized band structure of epitaxial graphene (red curve for up-spin, blue curve for down-spin). The Dirac point is situated 0.5 eV below Fermi energy; (**b**) Spin-polarized band structure of hydrogenated epitaxial graphene (red curve for up-spin, blue curve for down-spin). The arrows indicates the spin-splitted pseudo-Landau level *n* = 0.

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
