# Peer review of "Origin of Room-Temperature Ferromagnetism in Hydrogenated Epitaxial Graphene on Silicon Carbide"

_nanomaterials, 2019, doi:10.3390/nano9020228_

Round 1
Reviewer 1 Report
In their article, the Authors examine the origin of the ferromagnetism observed at room temperature in epitaxial graphene grown on silicon carbide. They conclude that it should be due to the fact that the presence of silicon dangling bonds, curving the graphene layer, should favour the hydrogenation of one graphene sublattice with respect to the other. The silicon dangling bonds further enhance ferromagnetism, due to Coulomb interactions which increase the magnetic moment.
In order to arrive at this conclusion, the Authors show existing results (reported also in Figs. 1-3 and 5) and new results that they numerically obtain by DFT (Figs. 4 and 6-8). In my opinion, a more clear distinction between the results already present in the literature and the new ones should be done (already in the introduction and then when the data are shown), in such a way as to stress the elements of novelty of the paper.
In the following, I list other points to clarify/improve.
1) line 16: "and the hydrogen bonding" -> "and by the hydrogen bonding"
2) line 25: since also papers with substitutional doping are cited, I suggest to add this reference:
P. Marconcini, A. Cresti, F. Triozon, G. Fiori, B. Biel, Y.-M. Niquet, M. Macucci, S. Roche, Atomistic Boron-Doped Graphene Field-Effect Transistors: A Route toward Unipolar Characteristics, ACS Nano 2012, 6, 7942-7947
3) line 34: "(all belongs to one sublattice) are bonded to hydrogen atom" -> "(all belonging to one sublattice) are bonded to an hydrogen atom"
4) line 65: "takes place by" -> "has been performed using"
5) line 71: "as it reveals a" -> "which is in"
6) line 72: in the symbol of Angstrom the ring over A is ill-positioned; in general, all around the paper many subscripts, superscript, etc are all-positioned
7) line 73: delete the red part
8) line 77: "represent" -> "present"
9) line 86: "observed" -> "and observed"
10) line 101: in "nm2" 2 is the exponent; moreover, itwould be better to clarify that the cross-section shows the structure orthogonally to the SiC surface
11) line 103: "Patterns with dark regions are separated" -> "Patterns appear, with dark regions separated"
12) lines 109 and 117: "Figure 2(a); therefore" -> "Figure 2(a). Therefore"
13) lines 112 and 114: after "R30" the symbol of degree is ill-positioned
14) line 113: "dashed line and full line" -> "a dashed line and a full line"
15) line 115: "by full line" -> "by a full line"
16) line 115: "section of" -> "section along"
17) line 117: "illustrates" -> "exhibits"
18) line 119: "Corrugations in graphene layer are smoother than that of buffer layer." -> "The corrugations in the graphene layer are smoother than those of the buffer layer."
19) lines 138 and 140: the symbol of degree is ill-positioned
20) line 140: "retained as that of a" -> "came back to the levels of"
21) line 142: "times of" -> "times"
22) line 146: "[36] which" -> "[36]. This"
23) line 149: "|B|; in" -> "|B|. In"
24) line 152: "standingding" -> "standing"
25) line 162: "G" should be a subscript
26) line 168: the Authors say that they consider a curved graphene layer, in order to partially include the effects of SiC and of the buffer layer; then they introduce H adatoms and they relax the structure. Could the authors detail on the numerical way in which they relax the structure, maintaining at the same time the curvature effect?
27) line 184: "symmetrically" -> "symmetrically located"
28) line 196: "change with the wave vector" -> "change with the variation of the wave vector"
29) line 198: the symbol of vector is ill-positioned
30) line 209: "atom to" -> "atom into"
31) line 219: "with a two hydrogen attachment" -> "with two hydrogen attachments"
32) line 224: "The preference for graphene for" -> "The preference for"
33) line 227: "resulted" -> "resulting"
34) line 234: "and, clearly seen," -> "and, as clearly seen,"
35) lines 235-236: "these spin polarized quasi-localized state connected regions" -> "these connected regions of spin polarized quasi-localized states"
36) line 239: "bandwidth of localized" -> "band of localized"
37) line 242: "leads to disappearing of" -> "leads to the disappearance of"
38) line 285: "used such that" -> "used, with"
39) line 295: "at K point" -> "at the K point"
40) line 307: "On the other hand," -> "Instead," (if I have understood well)
41) line 316: "besides to" -> "besides"
42) line 317: "These extra two" -> "These two extra"
43) line 326: with "simulated electron doping" do you refer to Hydrogen doping? This should be clarified in the text
44) line 330: "gives rise" -> "give rise"
45) lines 328-330: since this seems to be an important point, the Authors are encouraged to extend this sentence in order to further clarify the role of silicon dangling bonds
46) line 336: leading to favourable one sublattice site hydrogen occupation" -> "favouring one sublattice site hydrogen occupation"
47) line 340: "due the" -> "due to the"
Author Response
We thank the referee for his comments and suggestions for improvement of the paper.
a) In my opinion, a more clear distinction between the results already present in the literature and the new ones should be done (already in the introduction and then when the data are shown), in such a way as to stress the elements of novelty of the paper.
We add new text in the last paragraph of the introduction (line 53-60) to make our new theoretical results more explicit. Also in the beginning of paragraph 3.2 (line 161, 166) and in line (306) a more clear distinction between literature and our results is made.
b) All the textual suggestions for improvement have been implemented.
2) Reference
P. Marconcini, A. Cresti, F. Triozon, G. Fiori, B. Biel, Y.-M. Niquet, M. Macucci, S. Roche, Atomistic Boron-Doped Graphene Field-Effect Transistors: A Route toward Unipolar Characteristics, ACS Nano 2012, 6, 7942-7947 is added as ref.[16].
26) old line 168 has been changed to new lines 163-170.
43) old line 326, “with simulated electron doping”. This is already defined in point 26, lines 163-170 and in new line 340, “simulated effective electron doping”
45) old lines 328-330. This sentence has been removed in the new text lines 334.
Reviewer 2 Report
The paper deals with room-temperature ferromagnetism of hydrogenated epitaxial graphene on silicon carbide.
The authors show that in epitaxial graphene on SiC, the presence of the Si-dangling bonds creates preferential sites for hydrogen adsorption and that this leads to a preferred one sublattice carbon occupation of H atoms, which is necessary for ferromagnetic behaviour. The chemical modification of the graphene layer by H adatoms generates localized pseudo Landau levels. Extended spin-polarized electron states from the n=0 Landau level give rise to long-range ferromagnetic coupling.
The subject is timely and interesting. The paper is well written and the work seems correctly conducted. As a result of a simulation, the conclusions are difficult to control, but appear reasonable and are convincingly presented.
In my opinion, the paper is ready for the publication. Here are a few minor suggestions:
Magnify the figures. Figure 1, in particular, is difficult to see. Use a color different than white for the diamond cell lines.
Line 166: “According to the observations of Ref. [25], hydrogen atoms preferentially bond to sites where the lattice is maximally convexly curved, so we added one hydrogen atom to the highest carbon atom in the lattice unit cell of calculations.” For self-consistency, I suggest adding a sentence with the physical reason why hydrogen atoms preferentially bond to sites where the lattice is maximally convexly curved.
Line 233: “The attached hydrogen atoms (remarked by arrows in the figure)” The arrows are barely visible in the printed-paper: make them clearer.
Author Response
We thank the referee for his comments and suggestions for improvement of the paper.
Both figures Fig.1 and 6 have been improved according the suggestions of the referee.
1) Old Line 166 has been changed new line 177-179.
Round 2
Reviewer 1 Report
The article has been much improved and I suggest its publication in the journal.
I suggest only these further minor corrections:
1) line 28: "researches" -> "researchers"
2) line 103: "modulate (6x6) SiC incommensurate" -> "incommensurate (6x6) SiC modulation"
3) line 212: "clear noteworthy" -> "noteworthy"
4) line 285: "coincide" -> "coincides"
5) line 286: "layer), as" -> "layer); as"
Author Response
minor corrections as has been suggested by the reviewer have been implemented.